# The Role of Cerebellar Intrinsic Neuronal Excitability, Synaptic Plasticity, and Perineuronal Nets in Eyeblink Conditioning

**DOI:** 10.3390/biology13030200

**Published:** 2024-03-21

**Authors:** Bernard G. Schreurs, Deidre E. O’Dell, Desheng Wang

**Affiliations:** 1Department of Neuroscience, West Virginia University, Morgantown, WV 26505, USA; dswang@hsc.wvu.edu; 2Department of Biology, Earth and Environmental Sciences, Pennsylvania Western (PennWest) University, California, PA 15419, USA; odell@pennwest.edu

**Keywords:** cerebellum, classical conditioning, intrinsic membrane excitability, perineuronal nets, synaptic plasticity

## Abstract

**Simple Summary:**

Eyeblink conditioning is a simple form of learning that has been used to study areas of the brain involved in how we learn new tasks and how we remember them. One area of the brain that is important for eyeblink conditioning is the cerebellum. Changes that take place in the cerebellum involve a number of neural processes, including changes in the connections between neurons, changes in a neuron’s excitability, and even changes in the matrix that surrounds these neurons. Here, we explore these different processes and how they interact with each other to form the building blocks of a basic form of learning. Understanding how learning and memory take place may help us solve the mystery of how we lose the ability to learn and remember in diseases like Alzheimer’s disease, and how to we remember too much in post-traumatic stress disorder.

**Abstract:**

Evidence is strong that, in addition to fine motor control, there is an important role for the cerebellum in cognition and emotion. The deep nuclei of the mammalian cerebellum also contain the highest density of perineural nets—mesh-like structures that surround neurons—in the brain, and it appears there may be a connection between these nets and cognitive processes, particularly learning and memory. Here, we review how the cerebellum is involved in eyeblink conditioning—a particularly well-understood form of learning and memory—and focus on the role of perineuronal nets in intrinsic membrane excitability and synaptic plasticity that underlie eyeblink conditioning. We explore the development and role of perineuronal nets and the in vivo and in vitro evidence that manipulations of the perineuronal net in the deep cerebellar nuclei affect eyeblink conditioning. Together, these findings provide evidence of an important role for perineuronal net in learning and memory.

## 1. Introduction

There is a long history of studying learning and memory to understand how a large range of organisms, including humans, adapt to the demands of their environment. One particularly well-understood form of learning is classical conditioning, first described by Pavlov more than 100 years ago. The defining features of classical conditioning include the delivery of a relatively innocuous signal or warning followed, almost immediately, by a significant event. In the case of fear conditioning, a tone is usually followed by shock to the feet of a rat or mouse, or the fingers of a human. In the case of eyeblink conditioning, the same tone may be followed by a puff of air to the eye of a person, monkey, rabbit, rat, or mouse. The history of eyeblink conditioning began with the study of behavioral laws governing the acquisition, consolidation, and extinction of a conditioned response—closure of the eye during the tone and before the air puff. A growing interest in understanding not only the “how” but the “where” of eyeblink conditioning led to concerted efforts to search for the sites in the brain where learning and memory take place—the engram—and the particular role of the cerebellum. Although not universally accepted, there is evidence that the cerebellum is an engram for eyeblink conditioning. Research shows changes in synaptic and intrinsic membrane plasticity, as well as changes in perineuronal nets, are involved in the successful acquisition of eyeblink conditioning. We have previously reviewed evidence that eyeblink conditioning results in significant changes in intrinsic membrane excitability and synaptic plasticity in the cerebellum. Here, we review research showing that perineuronal nets surrounding principal neurons in the deep cerebellar nuclei (DCN) are involved in eyeblink conditioning and may mediate changes in intrinsic membrane excitability and synaptic plasticity.

## 2. Eyeblink Conditioning

Although head-fixed mice are the most recent research subjects to undergo eyeblink conditioning, Hilgard and colleagues reported eyeblink conditioning in the 1930s in dogs [1] and humans [2]. Considerable theoretical and experimental interest in human eyeblink conditioning continued into the 1960s [3], but conceptual and methodological issues dampened enthusiasm for human eyeblink research [4,5]. First, Gormezano and others raised conceptual and methodological concerns about the exclusion of subjects known as “voluntary responders”, who deliberately blinked to the tone or light during an eyeblink conditioning experiment [4]. Second, there was a growing frustration with an inability to analyze the brain of human subjects—sometimes considered a black box—to identify the biological basis of learning and memory. The latter limitation has since been overcome, in large part, by the use of imaging techniques, including positron emission tomography [6,7,8] and magnetic resonance imaging [9,10,11], that have identified the involvement of areas including the hippocampus, prefrontal cortex, and cerebellum in human eyeblink conditioning. These efforts to identify areas of the brain involved in human eyeblink conditioning were antedated in the 1980s by a shifting interest to search for the engram [12,13,14,15,16,17] in animal models of eyeblink conditioning [18].

The basic eyeblink conditioning paradigm involves presenting a conditioned stimulus (CS), such as a tone or light, that does not initially elicit an eyeblink response, and pairing it with an unconditioned stimulus (US), such as a puff of air or brief electrical pulse, near the eye that does elicit an eyeblink response (unconditioned response, UR). With repeated pairings of the two stimuli at specific durations and intervals, a conditioned eyeblink response (CR) can emerge. Importantly, explicitly unpaired presentations of the tone and air puff or shock to a different group of subjects to assess nonassociative contributors to responding, such as sensitization, show very low levels of responding to the CS [19,20,21]. In addition to the initial research with rabbits, interest in the use of rats to determine the neural substrates of eyeblink conditioning increased when Skelton overcame the limitations of restraint normally required for eyeblink conditioning in rats [22] by developing a procedure for recording eyelid EMG activity in freely moving animals [23,24,25,26,27,28]. Figure 1 shows an unpublished waterfall plot of responding in a freely moving adult rat given paired CS–US presentations. More recently, mouse eyeblink conditioning has been explored with several different recording techniques using head-fixed mice that walk on a movable surface, such as a ball or cylinder [29,30,31,32,33,34,35,36].

## 3. The Role of the Cerebellum in Eyeblink Conditioning

Early results in the search for the engram for eyeblink conditioning in rabbits documented the involvement of the hippocampus [37,38], and later findings described the more crucial role played by the cerebellum [39,40,41,42,43]. Lesion, inactivation, and recording studies extensively reviewed elsewhere [44,45,46,47,48,49,50] identified the deep cerebellar nuclei (DCN), particularly the anterior interpositus nucleus (AIN), as being important for eyeblink conditioning [36,50,51,52,53,54,55,56,57,58,59]. This was not a universally accepted position [48,60,61,62]. For example, in a consensus paper, Perciavalle et al. [63] provided evidence from rats, cats, and rabbits, among others, that the AIN participated in the timing and performance of ongoing conditioned eyeblinks but did not generate or initiate the conditioned response. Earlier, Welsh and Harvey showed that lesions of the rabbit AIN resulted in motor performance deficits in response timing and amplitude in both conditioned and unconditioned responses, but did not eliminate conditioned responses [60]. A similar debate has occurred about the role of the cerebellar cortex in eyeblink conditioning, particularly lobule HVI, which Yeo first showed was important in rabbit eyeblink conditioning [43,64,65]. As with the role of the AIN, the role of HVI in mice, rats, rabbits, and humans has been shown to be important [10,11,29,51,64,65,66,67,68] but not the sole site [69,70] for eyeblink conditioning. As with many controversies in the field, the devil is in the details, and the range of stimuli (e.g., tones versus lights, shock versus air puff), site and extent of the lesions (unilateral versus bilateral, reversible versus permanent), the conditioning parameters (interstimulus interval, intertrial interval, number of trials), method of restraint (head-fixed, cloth bag, solid restrainer, freely moving), modes of assessment (EMG, movement potentiometer, non-invasive reflectance, inductive coils, magnetic field effects), not to mention different species (mouse, rat, rabbit, cat, human) used to assess eyeblink conditioning, may all be important. All of these issues notwithstanding, research across these parameters, methods, and assessment modes with a very diverse range of species, including goldfish [71], turtles [72], rats [73], mice [74], guinea pigs [75,76,77], rabbits [41,60,70], and cats [78,79,80], as well as humans [8,81], suggests an important role for the cerebellum in eyeblink conditioning.

The question then becomes: what is the nature of the changes that take place in the cerebellum as a result of eyeblink conditioning? To understand the evidence, we first need to review the basic cerebellar circuitry underlying eyeblink conditioning. Figure 2 from [82] is a simplified illustration of some of the essential cerebellar circuitry involved in eyeblink conditioning. The figure shows that sensory inputs from the tone give rise to mossy fibers (MF) that synapse onto neurons in the AIN, and onto granule cells that give rise to parallel fibers that synapse onto Purkinje cell dendrites. Sensory inputs from the air puff give rise to climbing fibers (CF) from the inferior olive that also synapse onto neurons in the AIN, and then encircle the dendrites of a Purkinje cell.

Evidence of eyeblink-specific changes in the rabbit AIN was first reported as altered extracellular activity recorded with metal electrodes [39,41], and later also reported in the rat AIN [83]. Subsequently, evidence of conditioning-specific changes in the AIN began to emerge from electron microscopy, but those results remain somewhat confusing. On the one hand, there were no overall changes in synaptic number, shape, or perforations in excitatory neurons in the AIN of the DCN, although there was a significant increase in the length of excitatory synapses after rabbit eyeblink conditioning [84]. No distinction was made between mossy fibers and climbing fiber inputs. On the other hand, there was an increase in the number of excitatory synapses, but not inhibitory synapses in the rat DCN after eyeblink conditioning [85]. More recent work by Broersen et al. [86] has expanded on the Kleim et al. [85] study by showing an increase in the number of mossy fiber excitatory terminals, identified by VGlut1 labeling, in presumed projection neurons in the AIN of the mouse DCN as a result of eyeblink conditioning. In contrast to the Kleim study, Broersen et al. also showed an increase in the number of Purkinje cell inhibitory terminals to presumed AIN projection neurons as a result of eyeblink conditioning in the mouse. If the DCN is so important to eyeblink conditioning, it is difficult to reconcile these differences by simply pointing to the different species (rabbit versus rat versus mouse) or the different eyeblink conditioning procedures (tone versus light, air puff versus shock) that were employed. Perhaps it was an improvement in the sophistication of the techniques used to make the measurements. On the other hand, the changes may not have occurred exclusively in neurons involved in eyelid conditioning. For example, in addition to eyeblinks, Broersen et al. noted whole-body movements evoked by the conditioned stimulus. This idea may be supported by our finding that recordings from projection neurons in the juvenile rat AIN, labeled with a pseudorabies transsynaptic virus injected into the eyelid, showed increases in intrinsic membrane excitability following eyeblink conditioning in both labeled and unlabeled neurons [87]. These increases in excitability were significantly higher than in rats given unpaired presentations of the tone and shock.

## 4. Intrinsic Membrane Excitability Involved in Eyeblink Conditioning

A substantial amount of work has been performed to understand changes in intrinsic membrane excitability—changes in the likelihood of generating an action potential—and how these changes are involved in eyeblink conditioning. Changes in intrinsic membrane excitability are changes in the movement of ions across the membrane and come about by the insertion, removal, or modification of sodium, calcium, potassium, or non-selective cation channels [88,89,90,91,92,93,94,95,96,97,98,99,100,101,102,103,104,105,106,107,108,109,110,111]. These changes can occur as an increase or decrease in input resistance, membrane potential, afterhyperpolarization, spike threshold, spike frequency, or a combination of these properties. Intrinsic membrane excitability changes—reviewed extensively elsewhere [44,99,112,113,114,115,116,117]—have been found across a range of structures, species, and learning paradigms. Of relevance to this review are the experience-dependent changes in membrane properties found in the cerebellum [34,44,87,88,90,95,101,111,114,117,118,119,120,121,122,123,124,125], particularly eyeblink conditioning [34,44,87,126]. In a series of mouse eyeblink conditioning experiments, Titley, Hansel and colleagues examined changes in Purkinje cell membrane excitability [34] and found that increases in excitability were mediated by a Purkinje cell-specific calcium-activated K+ channel (SK2) [126]. We have also reported learning-specific changes in intrinsic membrane excitability in Purkinje cell dendrites, but as a function of rabbit eyeblink conditioning [127,128,129]. We were able to show that 4-AP—which blocks a transient voltage-dependent potassium channel at low concentrations—reduced afterhyperpolarization-mediated changes in excitability [127,130] and concluded that the observed eyeblink conditioning-specific increase in dendritic excitability was a function of changes in an I_A_-like potassium current—a finding consistent with changes in hippocampal CA1 pyramidal cells and interneurons as a function of eyeblink conditioning in the rabbit [131,132]. We have also reported an increase in intrinsic membrane excitability, measured as a reduction in the afterhyperpolarization of projection neurons in the AIN as a function of eyeblink conditioning in juvenile rats [87]. Importantly, dendritic excitability has been shown to be increased by the trafficking of voltage-dependent potassium channels [92].

## 5. Synaptic Plasticity in the Cerebellum Involved in Eyeblink Conditioning

Synaptic plasticity in the cerebellar cortex takes the form of long-term depression (LTD) and long-term potentiation (LTP)—phenomena that have been reviewed elsewhere [133,134,135,136,137,138,139]. LTD and LTP in the deep cerebellar nuclei have also been studied extensively [140,141,142,143,144], particularly in slice work by Raman and colleagues [145,146,147,148,149]. In the cerebellar cortex, stimulating climbing fibers and parallel fibers that synapse onto Purkinje cells reduces parallel fiber synaptic potentials—first proposed by Marr [150] and Albus [151] and later confirmed by Ito [152,153] and Gilbert and Thach [154]—has become a benchmark for studying the molecular underpinnings of synaptic plasticity. In the DCN, stimulation of white matter, which activates mossy fiber collaterals, has been shown to result in LTP of large, fast-spiking excitatory neurons, but only in the presence of hyperpolarization that mimics Purkinje cell inhibition [147,155]. Stimulation of Purkinje cell inhibitory inputs to these same large, fast-spiking excitatory neurons has been shown to result in either LTP or LTD depending on the stimulation parameters [143,144,156,157,158].

Although Ito first used cerebellar LTD to explain experience-dependent adaptation of the vestibulo-ocular reflex [159], it has also been used to explain eyeblink conditioning [138,160,161,162,163,164,165]. As we saw in Figure 2, information about the tone and air puff used in eyeblink conditioning reaches the cerebellum through mossy fibers and climbing fibers that synapse at the DCN and Purkinje cells [166,167,168]. In the case of delay eyeblink conditioning, where the tone and air puff overlap and co-terminate, activation of parallel and climbing fibers occurs together during the overlap, which could result in long-term depression of Purkinje cell inhibition of the DCN and long-term potentiation of mossy fiber excitation. Significantly, Broersen et al. [86] found an increase in the number of Purkinje cell inhibitory terminals and mossy fiber excitatory terminals in the AIN of the mouse DCN that may mediate LTD and LTP, respectively, as a result of eyeblink conditioning. As with the contrasting views about the role of the cerebellum in eyeblink conditioning described above, there are also contrasting views on the role of cerebellar synaptic plasticity in eyeblink conditioning. Some argue that it is necessary for eyeblink conditioning [169,170,171,172], and others have shown that eyeblink conditioning can occur in the absence of synaptic plasticity [161,173,174,175]. There are also more nuanced positions suggesting that synaptic plasticity is one of several forms of plasticity involved in eyeblink conditioning [44,114,117,174].

Although much of the debate about the role of synaptic plasticity in eyeblink conditioning has been based on delay conditioning where stimuli overlap, eyeblink conditioning in many species, including rats, rabbits, and humans, can be achieved with a trace conditioning paradigm where the tone ends 250–500 milliseconds before the air puff, creating a gap or “trace” between the stimuli [28,176,177,178,179,180,181,182,183]. As reviewed elsewhere [44], delays in transmission along the mossy/parallel fiber circuit have been proposed that ensure sensory inputs from the tone and air puff arrive in the cerebellum at the same time [184]. However, auditory transmission rates [185] and direct electrical stimulation that substitutes for tone and air puff all suggest there is no significant delay in activation of mossy fibers or climbing fibers [186]. Others have found that the medial prefrontal cortex may bridge the trace and then activate mossy fibers that are contiguous with a climbing fiber input [187,188,189]. This is, in essence, a serial compound of two sequential mossy fiber inputs (directly from the brainstem auditory system and then indirectly from the prefrontal cortex) followed by a climbing fiber input. Kehoe and others have explored serial compound conditioning extensively, and there is very good evidence that serial compounds of two stimuli allow animals to bridge temporal gaps of the order of seconds—gaps that are otherwise too long to support eyeblink conditioning [190,191,192,193,194]. However, behavioral studies of serial compounds have invariably used two different stimuli—usually a tone and a light—and it is not clear that if the same stimuli were presented serially, the temporal gap would be bridged. When two identical stimuli have been used in sequence, rabbit and rat eyelid conditioning has been achieved, but the stimuli were both periorbital shocks that elicited an eyelid response [195,196]. It might be too simple to suggest that these shocks only elicited climbing fiber inputs to the cerebellum, although evidence suggests that climbing fiber activation can induce plasticity [197,198]. Shock has been described as an “inadequate” stimulus because direct electrical stimulation of nerve fibers in the region are qualitatively different from the sensory transduction that is required to detect light, sound, and touch. Nevertheless, shock elicits sensations like buzzing and pain that could reach the cerebellum along mossy fibers. In other words, presentation of two serial shocks would consist of both MF and CF inputs to the cerebellum that could alter synaptic properties leading to eyeblink conditioning [195].

## 6. Membrane Excitability and Synaptic Plasticity in Eyeblink Conditioning

The question is: how do distant changes in synaptic plasticity at specific dendrites translate to changes in the output of the cell [14,44,93,99,113,114,123,199]? This is particularly true in Purkinje cells where the dendritic tree is extensive, and there are thousands of synaptic connections from parallel fibers, as well as a smaller number of inhibitory connections from molecular layer interneurons. If, as documented elsewhere [44,127,129], there are both membrane and synaptic conditioning-specific changes following eyeblink conditioning, what is the sequence of those changes? Does synaptic plasticity occur first, followed by membrane plasticity, or vice versa? If membrane excitability increases occur first, then any changes at specific synapses in the dendrites—which would be considered to be more subtle because of the size of the dendritic tree—would presumably be amplified. Conversely, if synaptic plasticity changes occurred first, they would have to wait for changes in membrane excitability to be effective in altering a neuron’s output. What does this mean for behavior? If cells in the deep cerebellar nuclei are more excitable, the same synaptic input from MFs would cause AIN neurons, which initially only fired to the input provided by both the MFs and CF collaterals, to fire more readily, leading to an increased likelihood of a CR when only the MF collaterals have input to the AIN, which is what we see on tone-alone test trials. This would then be amplified by synaptic plasticity—the decreased inhibition of deep cerebellar nuclei from Purkinje cell axons and increased excitation from mossy fiber inputs already described above [86]. Although the evidence for the involvement of both synaptic plasticity and membrane excitability in eyeblink conditioning reviewed so far is persuasive, this may only be part of the story, because there is increasing evidence that the perineuronal nets surrounding the deep cerebellar nuclei may also play a role in eyeblink conditioning [200,201,202,203].

## 7. The Perineuronal Net

Cajal may have been the first to observe the perineuronal net (PNN) in the cerebellum, but Golgi appreciated that the PNN was more than a tissue processing artifact and is credited with identifying the PNN as a structure and describing it in detail [204,205]. The original concept of communication in the brain comprising synaptic connections between neurons at a pre- and post-synaptic interface, first proposed by Cajal [168,206], has since been expanded to include the vital role of glia that together form the tripartite synapse [207,208,209]. More recently, the importance of the PNN surrounding the soma and proximal dendrites of a neuron, and being actively involved in modulating communication, has prompted the notion of a tetrapartite structure [210,211,212,213].

The PNN is a form of extracellular matrix that forms a reticular structure around several different classes of neurons in the brain, particularly fast-spiking neurons [204,214,215,216,217,218,219,220], including projection neurons in the DCN that can fire at rates in excess of 100 Hz [221,222]. In fact, the PNN is found covering more cells in the DCN than any other part of the brain [223,224]. The PNN is composed of hyaluronic acid, link proteins, chondroitin sulfate proteoglycans (CSPGs), and tenascin-R that assemble into a dense, lattice-like sheet that can be disrupted by chondroitinase ABC (ChABC), an enzyme that degrades the glycosaminoglycan side chains of chondroitin sulfate proteoglycans. Development of the PNN has correlated with critical periods of plasticity [225,226,227,228,229,230]. The Bruckner group identified stages of PNN maturation across different regions of the postnatal rat brain from P0 [231] as well as by Ye and Miao, who studied PNN development in the postnatal mouse visual cortex from P10 to P42 [232]. We have reported development of the PNN in the rat DCN, where the PNN does not fully assemble into its lattice-like structure until the end of the third week of post-natal development [201]. Figure 3, adapted from [201], shows the development of the PNN in the rat DCN—determined by the labeling of CSPGs with WFA (*Wisteria floribunda* agglutinin). The figure shows that the PNN surrounding neurons in the DCN does not fully develop in rat pups until after post-natal day 18 (P18). Interestingly, this is also around the time when electrical properties of neurons in the rat DCN mature with increases in the amplitude of the afterhyperpolarization, a prolonged interval between the first and second evoked action potential, and an increase in afterhyperpolarization amplitude for hyperpolarization-induced rebound spikes [233]. This is also when climbing fiber pruning nears completion [234] and rats are first able to acquire eyeblink conditioning to either tones or lights paired with shock [201,235,236,237,238,239], although they can acquire conditioned responses at early time points if other stimuli are used, including two shocks [196] or direct brain stimulation [240].

The role of the PNN in eyeblink conditioning has been explored by several groups [200,201,202,203]. Hirono et al. used enzymatic digestion of the PNN with chondroitinase ABC (ChABC) in cerebellar slices of the DCN and found a decrease in Purkinje cell inhibitory postsynaptic currents, as well as higher terminal levels of eyeblink conditioning in head-fixed mice treated with ChABC compared to mice infused with the vehicle [203]. Carulli et al. used a lentiviral approach to release ChABC into the mouse DCN and showed a reduction in spontaneous activity of DCN neurons that may have been due to increased Purkinje cell inhibitory inputs and decreased mossy fiber excitatory inputs. They suggested that this could explain the enhanced plasticity in the DCN during the acquisition of eyeblink conditioning [202,241]. We recently showed that in vivo degradation of the PNN by ChABC using indwelling cannulae resulted in significant reductions in freely moving rat eyeblink conditioning amplitude and area compared to saline-infused controls, but did not affect conditioned or unconditioned response frequency [201]. Figure 4 shows an example of PNN digestion in the left DCN (ChABC) compared to a control infusion on the right (saline) four days following infusion. Although there was a 50% reduction in the percentage of neurons with WFA labeling, digestion in the DCN was not complete, and there were still neurons with WFA reactivity throughout the structure. The remaining neurons with WFA labeling may explain why there were significant changes in the amplitude and area of CRs without a reduction in the frequency of responding. We next conducted an in vitro experiment in which slices of the cerebellum were incubated with ChABC, and found the AIN had fewer WFA-positive neurons (41.98% ± 4.75) compared to the AIN in slices incubated with the vehicle (98.71% ± 0.38), *p* < 0.001. Neurons exposed to ChABC required more current to fire an action potential (AP) and had a longer latency to evoke an AP compared to cells in the vehicle group. AIN neurons exposed to ChABC also showed a longer inter-spike interval and had a larger afterhyperpolarization amplitude, shown in Figure 5. There also appeared to be a more robust digestion in our in vitro condition compared to our in vivo study. Interestingly, there were no differences in the membrane potential or input resistance. These results suggest that digestion of PNN with ChABC in acute AIN slices decreased the intrinsic excitability of large excitatory neurons without affecting other membrane properties. Although we saw decreased excitability in fast-spiking projection neurons in DCN in vitro, Hayani et al. saw no changes in mouse hippocampal fast-spiking interneurons or principal cells after treatment with ChABC in vitro [242]. In a comprehensive review of electrophysiological consequences of PNN modification, Wingert and Sorg (2018) concluded that removing PNNs has an impact on the synaptic and membrane properties of fast-spiking interneurons, but less so on principal neurons [241,242].

There has been considerable discussion about the role of the PNN in synaptic plasticity. In examining perineuronal nets of the adult rat cerebellum, Carulli et al. stated that PNNs have “holes at the sites of synaptic contacts” [243]. The concept of holes for synaptic contacts onto neurons in rat deep cerebellar nuclei was suggested by Lafraga et al., who noted the PNN had “holes for the synaptic boutons.” [244]—an idea proposed much earlier by Schwartz [245]. Tsien reflected on the function of the PNN by suggesting long-term memories were stored in the pattern made by these holes [246]. In a recent review, Rudolph et al. noted PNNs restrict new synapses from being produced and old synapses from being pruned, which is thought to regulate neuronal plasticity [247]. This view was included among those described by Celio et al., who also provided evidence that the PNN may maintain cellular relationships, concentrate growth factors, generate an ion-buffering microenvironment, prevent extracellular space occlusion, and form a link with the intracellular cytoskeleton [205]. Carulli et al. suggested that PNNs are strategically positioned to influence the development and stabilization of synaptic connections [202]. Frischknecht et al. showed that PNN removal facilitated AMPA receptor movement across the membrane, whereas NMDA receptors did not move with PNN removal [248].

There has been much less discussion about the role of the PNN in intrinsic membrane excitability. Theorizing about the function of the PNN includes regulating the localization of ion channels [249,250], binding cations [251], gating ion channels [252], anchoring ion channels, ion exchangers, and ion transporters in the plasma membrane, as well as reducing membrane capacitance by acting as an electrostatic insulator [253]. PNNs may act as local buffers of sodium and potassium ions in the extracellular space to ensure rapid ion transport [254,255]. In an extension of that idea, Morawski et al. suggested the PNN contains anionic binding sites that trap extracellular calcium, potassium, and sodium that can be mobilized in the service of the demands of fast-spiking neurons [256]. If PNNs act as an anchor for ion channels, it follows that disrupting the PNN may make ion channels more able to be inserted or removed. This could explain the decreases in membrane excitability we observed by disrupting the PNN with ChABC—an increase in voltage-dependent potassium channels in the membrane. In measuring the effects of the PNN on membrane properties, Frischknecht et al. showed that PNN removal did not affect resting potential, AP amplitude, or width, but they did not measure other indices of membrane excitability [248]. We also did not see changes in resting membrane potential or action potential amplitude as a function of treating the PNN with ChABC, but we did observe alterations in other measures of membrane excitability, particularly the amplitude of the afterhyperpolarization [201], which may have resulted from the insertion of voltage-dependent potassium channels.

Taken together, the data suggest the PNN may control cellular and synaptic forms of plasticity by regulating the localization of ion channels, particularly potassium channels, and receptors, particularly AMPA receptors [92,241,249]. It has even been suggested that these functions may be the result of modification of the PNN by microglia [225] through the manipulation of proteases and phagocytosis [212]. The absence of local microglia through experimental depletion has been shown to enhance PNN deposition and density, in addition to affecting synaptic number [212]. As we have noted, insights into the role of the PNN have also come from direct enzymatic and genetic manipulation of the PNN. A recent review by Fawcett et al. described the effects of modulating synaptic function by genetically or enzymatically perturbing the PNN and the potential effects on a large range of learning and memory, including eyeblink conditioning [257]. We suggest that perturbing the PNN may also have consequences for intrinsic membrane excitability—another form of plasticity that underlies eyeblink conditioning.

## 8. Conclusions

Eyeblink conditioning results in significant changes in intrinsic membrane excitability and synaptic plasticity in the cerebellum in a number of species. As in other preparations, there is a growing consensus that both intrinsic membrane excitability and synaptic plasticity are required for eyeblink conditioning [88,99,101,114,117,121,122,123,131,132,199,258,259,260,261,262,263]. More recently, there is a growing consensus that perineuronal nets are also involved in learning and memory across a range of paradigms [210,264,265,266,267,268,269,270], including fear conditioning [271,272,273,274,275,276,277,278] and eyeblink conditioning [86,200,201,202,203,257]. Together, these findings provide evidence for the combined role of intrinsic membrane excitability, synaptic plasticity, and the perineuronal net in eyeblink conditioning. Evidence is growing that the perineuronal net may alter learning and memory by regulating intrinsic membrane excitability and synaptic plasticity.

## Figures and Tables

**Figure 1 biology-13-00200-f001:**
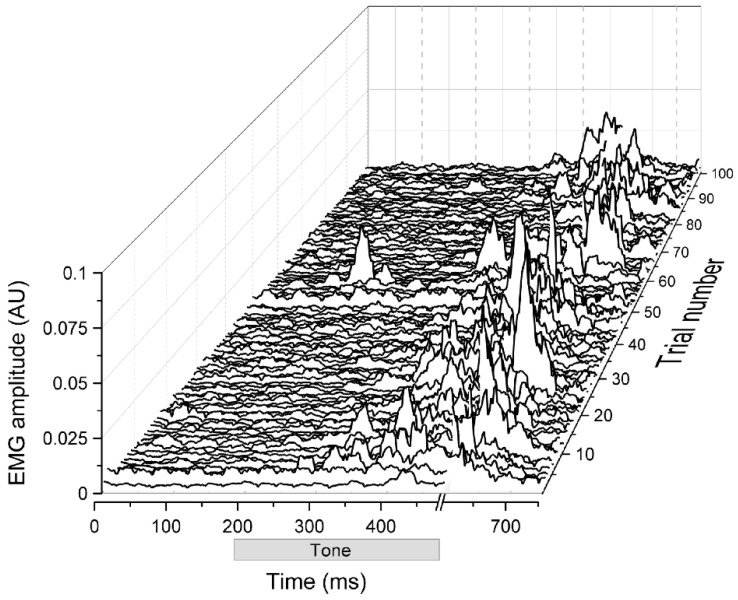
Rat eyeblink conditioning. A waterfall plot of eyeblink responding in a rat shown as filtered and rectified EMG signals (in arbitrary units) across 100 (Trial Number) stimulus presentations of a 380-ms tone conditioned stimulus (CS, gray rectangle starting 200 ms from trial onset) paired 90 times with a 100-ms periorbital shock unconditioned stimulus (US). The tone CS was presented alone every tenth trial. The EMG signal was blanked at the break in the x axis to prevent the low-voltage EMG signal from being swamped by the larger voltage of the shock, which would otherwise have produced ringing in the EMG amplifier that would have outlasted the response. Part of the unconditioned response can be seen after the break in the EMG signal. Unpublished data.

**Figure 2 biology-13-00200-f002:**
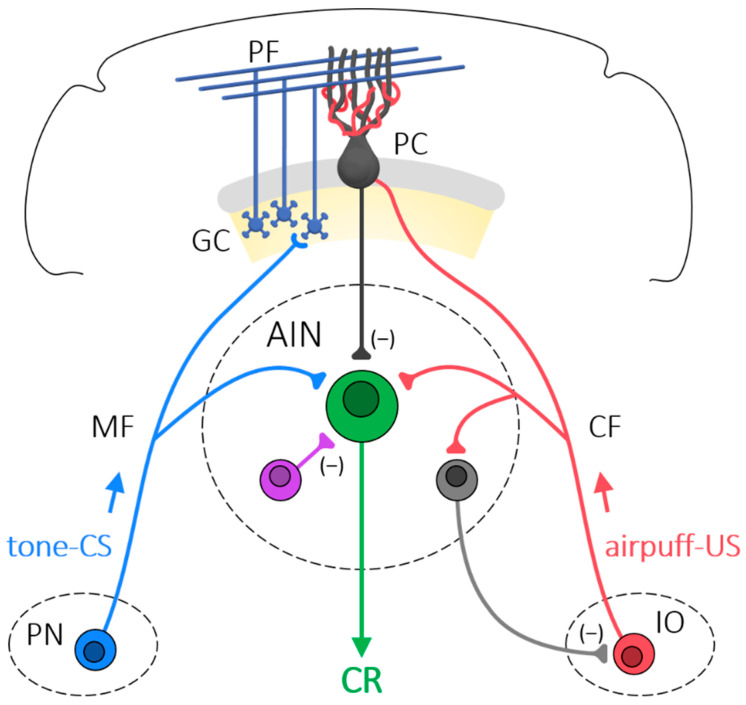
Simplified cerebellar circuit involved in eyeblink conditioning. Auditory information from the tone conditioned stimulus (CS) shown in blue travels along mossy fibers (MF) from the pontine nuclei (PN). MFs send collateral fibers to neurons in the anterior interpositus nucleus (AIN) of the deep cerebellar nuclei, and then travel up to the cerebellar cortex and synapse onto granule cells (GC). Granule cell axons then bifurcate and synapse on the dendrites of Purkinje cells (PC). There may be thousands of PF synapses onto one PC. Sensory information from the air puff unconditioned stimulus (US) is shown in red travels from the inferior olive (IO) as climbing fibers (CF) that also send collaterals to the AIN, and then travel up to the cerebellar cortex to wrap around and synapse onto the dendritic tree of a Purkinje cell. There is typically only one climbing fiber for each PC. MF and CF synapses are excitatory and PC output to the AIN is inhibitory (−), as are interneurons in the AIN. Figure from [82].

**Figure 3 biology-13-00200-f003:**
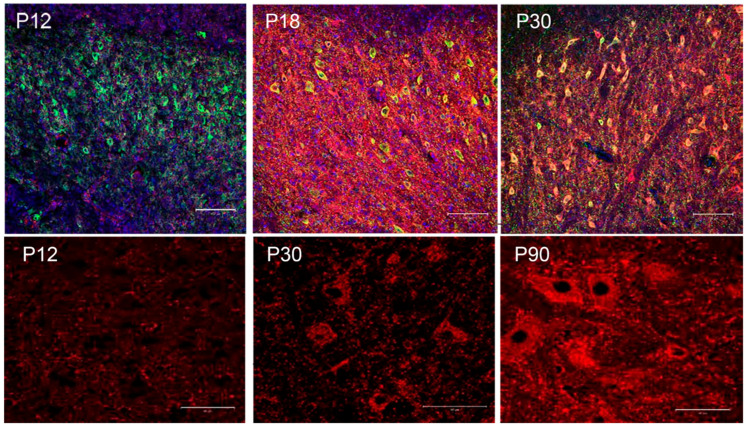
Development of the perineuronal net in the anterior interpositus nucleus of the rat cerebellum. The top panels show WFA reactivity (red), DAPI (4′,6-diamidino-2-phenylindole, blue), and MAP2 (microtubule-associated protein 2, green) reactivity in the rat AIN at P12 (A), P18 (B), and P30 (C) at 20×. Scale bars = 100 μm. The bottom panels show an increase in WFA reactivity (red) alone at P12, P30, and P90 at 63×. Scale bars = 50 μm. Figure modified from O’Dell et al. [201].

**Figure 4 biology-13-00200-f004:**
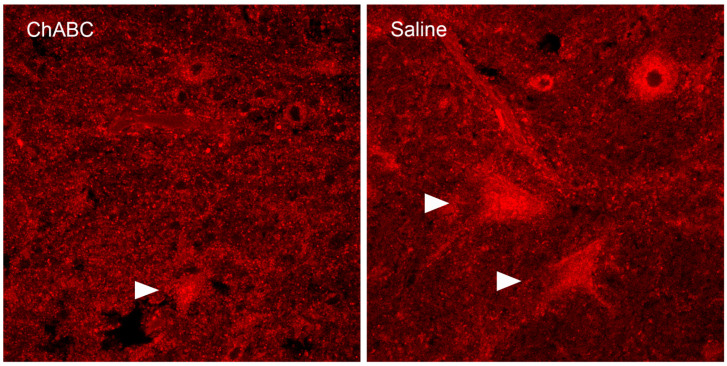
Disruption of the PNN in the DCN with ChABC. The left panel shows WFA labeling in the left anterior interpositus of the deep cerebellar nuclei four days after the rat received an infusion of the enzyme chondroitinase ABC (ChABC). The right panel shows the right anterior interpositus nucleus in the same rat that received an infusion of saline. ChABC was found to have reduced the number of WFA-labeled neurons (42.38 ± 5.24%) compared to the side receiving the vehicle (68.78 ± 5.14%), *p* < 0.0001. Arrowheads indicate WFA-labeled neurons.

**Figure 5 biology-13-00200-f005:**
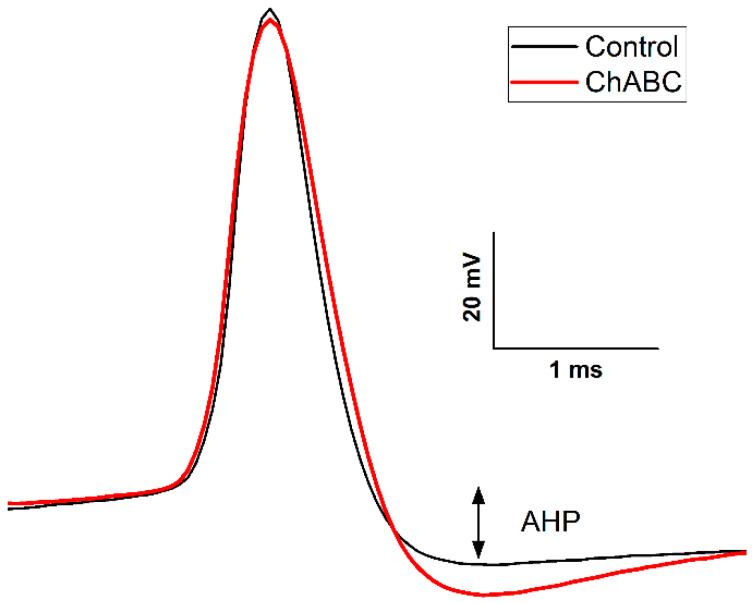
Increase in afterhyperpolarization of a neuron in the DCN after incubation in ChABC. An action potential recorded in a principal neuron of the rat AIN following incubation of a slice of the cerebellum in chondroitinase ABC (ChABC). Slices were incubated for 8 h in either a ChABC concentration of 0.25 U/mL (red trace, ChABC) or 250 μL of the 0.01% bovine serum albumin solution added to the ACSF as a vehicle (black trace, Control). The figure shows that after ChABC incubation, the size of the afterhyperpolarization (AHP) was significantly larger. The statistical results and methodological details are reported in [201].

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
