# Peer review of "The Role of Cerebellar Intrinsic Neuronal Excitability, Synaptic Plasticity, and Perineuronal Nets in Eyeblink Conditioning"

_biology, 2024, doi:10.3390/biology13030200_

Round 1

Reviewer 1 Report

Comments and Suggestions for Authors

The current work by Schreurs et al provides a review of the role of the cerebellum and the different types of plasticity in eyeblink conditioning.

Below are some comments that need to be addressed before the manuscript can be considered for publication:

1. Is the citation for Gonzalez-Joekes (2014) in Figure legend 1 and 2 correct? I could not locate the citation in the references.

2. Figure 2A and B, what do the numbers in the Y-axis mean?

3.  In Figure 3, the X axis is missing and the Y-axis is again not specified.

4.  In line 84, the authors note that the role of the cerebellum in eyeblink conditioning is not universally accepted. Since the current review work looks at the role of the cerebellum in eyeblink conditioning, they authors should expand on these critics a bit more.

5. The order of Figure 6 and Figure 7 are flipped. For clarity, it should be fixed.

6.  Line 149 should include the citation.

7.  Is the Wang et al 2018 citation in Figure 7 correct? I could not find the figure in the original article.

8. Citation no 38 should not be included in line 235 since it is a review article.

9.  It is unclear if Figure 9 is new original data being presented in this article or adapted from previous work.

10. Are the citations in Figure 4 legend correct? I could not find them in the references.

Reviewer 2 Report

Comments and Suggestions for Authors

The authors reviewed the recent progress on the role of intrinsic membrane excitability, synaptic plasticity and perineuronal nets in eyeblink conditioning. However, this review requires some extensive revision before publication.

Major revision:

1.      The authors discuss extensively their own findings regarding this topic. For part 4 to part 8, the author extensively described their own series of findings. For a comprehensive and systematic review, the authors should also discuss the relevant work from other people in the field and compare those with their own findings.

2.      In the discussion of intrinsic membrane excitability (the paragraph starting line 162), the authors described a series of findings related to the phenomena that involve modifications of ion channels underlying membrane excitability changes. I do not see the relationship between some described results with the later discussion of the intrinsic membrane property changes observed in eye blinking. For example, how is morphine withdrawal mentioned in line 167 related?

3.      More importantly, I expect to see some synthesis of findings at each part. For example, in discussing dendritic spikes for eyeblink conditioning, how does the channel property of potassium channel IA – described by the author – affect the computation of the neural circuit? What is the biological meaning of PKC pathway in this aspect? This extends to the discussion of synaptic plasticity, where the authors listed all the changes they observed (again, it needs to be balanced with other’s findings in the field). However, how do the described changes affect the neural circuit computation? Moreover, the synthesis of membrane excitability and synaptic plasticity in part 7, the authors mainly discuss the question of which comes first – membrane plasticity or synaptic plasticity? It could be extended to discuss other aspects of the synthesis. How do we integrate the recent findings on membrane and synaptic plasticity to understand the neural circuit for such behavior.

Minor revision:

1.      When authors discuss DCN in line 100, they could provide some brief explanation on the DCN circuit.

2.      Figure 2, in day 1, I believe it should be 40 for line 2 rather than 30.

3.      Figure 5, Panel B instead of Pane B

Comments on the Quality of English Language

Some typos need to be corrected.

Reviewer 3 Report

Comments and Suggestions for Authors

This is a well written review of the relationship between neural intrinsic properties in the cerebellum and eyeblink conditioning from an author who has been extraordinarily prolific in the field. However, 75% of the topics covered in this manuscript have already been reviewed by the same author in 2019. The current text is just different enough not to call the manuscript self-plagiarism, but it is close (take a look at the first sentence of the introduction and the first sentence of the abstract of the 2019 review). The majority of the paper is a review of the authors’ own work, rather than a perspective on the current state of the field. The only novel part of the manuscript is the one considering perineuronal nets (PNN). The manuscript quite carefully notes that the link between PNN, intrinsic properties and eyeblink conditioning behavioral results is highly speculative, this should be evident in the Abstract as well. My recommendation would be to drop all the material already reviewed in Schreurs 2019 and focus on the potential role of the PNN, it may be a bit provocative, but it would provide solid reading and interesting, fresh ideas, instead of rehashing the authors’ previous work another time.

Major issues

1. The Abstract is all over the place. It makes strong statements that are very general, (e.g. cerebellum is important for cognition and emotion or the suggestion that perineuronal nets are causal for such functions) without much backing or even relevance. The abstract should be fact based and focus on the goals of the review rather than lofty claims. The Abstract also announces the goals of the review twice, this section must be more concise and better structured. Finally, the abstract claims to provide evidence for the PNN - intrinsic properties - eyeblink conditioning behavior relationship, while the main text is clearly stating that such relationship is completely speculative.

2. Recycled figures: Figures 1, 2 and 4 are reused figures from Gonzalez-Joekes J (2014) doctoral thesis and the Schreurs 2019 paper that is already a reused figure from the aforementioned publication. How many times can an author recycle their own figures?

3. Inappropriate citations: The intrinsic membrane properties section is littered with tangential or irrelevant citations. These include some blatant self-citation by the authors (sometimes from removed, irrelevant fields). Citations for this section should be reviewed and the whole section should be re-written to match the topic of the review: cerebellum, learning and memory, conditioning, etc. Here are some examples (not complete list): Line 167: how is reference 114 and this sentence related to the topic of the review? If the authors want to show an example, find one within the reviewed field. Line 176: how is reference 117 and this sentence related to the topic of the review? Use relevant citations. Line 180: self-citation referencing the author’s work in Alzheimer's disease, not related to the review topic.  Line 179-182: why use AD references?

Minor points

- Line 15: what is “membrane plasticity”? Do the authors refer to the changing nature of lipid bilayers? If so, how is that connected to synaptic plasticity? Or is this just a typo?

- Line 38 is essentially repeating the statement from line 30

- Citations in figures should match citation style in the manuscript and should also be cited in the main text, not only in the figure legend.

- There are many, sometimes redundant figures showcasing eyeblink conditioning behavior, the related anatomy and then perineuronal nets, but not a single illustration depicts intrinsic neuronal properties or plasticity, the main focus of the review, this needs to be remedied.

- Line 154: membrane plasticity is NOT synonymous with intrinsic excitability or any of the phrases in that sentence.

Round 2

Reviewer 1 Report

Comments and Suggestions for Authors

The authors have addressed the concerns that were raised in the initial review and expanded the discussions in the relevant sections. The manuscript can be accepted for publication. 

Reviewer 2 Report

Comments and Suggestions for Authors

I appreciate the authors' extensive rewriting of the review. I think it is good for publication now. However, at least in the pdf format I received, the layout of Figure 4 and Figure 5 is overlapping. But this should be a minor issue that can be fixed in the publication process.

Reviewer 3 Report

Comments and Suggestions for Authors

The authors revised their manuscript responding to and fixing all issues I raised.